# Effect of Fiber Reinforcement on Creep and Recovery Behavior of Cement–Emulsified Asphalt Binder

**DOI:** 10.3390/ma15217451

**Published:** 2022-10-24

**Authors:** Xiantao Qin, Siyue Zhu, Rong Luo

**Affiliations:** 1School of Transportation and Logistics Engineering, Wuhan University of Technology, 1178 Heping Avenue, Wuhan 430063, China; 2School of Civil Engineering and Architecture, Wuhan Polytechnic University, 68 Xuefu South Road, Wuhan 430023, China

**Keywords:** fiber reinforced cement–emulsified asphalt binder, repeated creep recovery test, creep recovery ratio, accumulated strain, creep compliance

## Abstract

In order to evaluate and improve the deformation behavior of cement–emulsified asphalt binder (CA) in cement–emulsified asphalt mixture, this study investigated the reinforcement of small additions of fibers (2%, 4%, and 6% addition by mass of cement) on the deformation resistance of CA. A repeated creep recovery test was implemented that measures the recovery rate of creep deformation and accumulated strain. Further, an improved piecewise curve-fitting method was used to determine the parameters of Burgers model, then the creep compliances were fitted and calculated. The results show the repeated creep recovery test to be a suitable method for obtaining useful information about creep and recovery deformation of fiber-reinforced CA. The influence of fiber types and dosages on the deformation recovery ability is determined based on the creep recovery ratio and accumulated strain. The improved piecewise curve-fitting method has high accuracy. Thereafter, the reinforcement effect was analyzed through the evolution of creep compliance under loading. Therefore, this paper can provide a reference for enhancing the properties of cement–emulsified asphalt mixture by maximizing the fiber reinforcement.

## 1. Introduction

Cold-mix asphalt mixture (CMA) has become a hot research topic within the background of carbon peak and neutrality goals because of its advantages to energy conservation and emission reduction and because of its easy construction. Cement–emulsified asphalt mixture was developed soon after CMA in order to enhance the adhesion between asphalt and aggregates, which in turn leads to an increase in mechanical properties [1]. In recent years, traditional research has most frequently focused on the material composition, gradation design, and long-term road performance of CMA, all of which greatly promote the development of CMA [2,3,4]; microwave radiation, especially, can improve CMA’s early strength [5].

Then, the focus of CMA shifted from macroscopic to microscopic by using X-ray diffraction, scanning electron microscope (SEM), and environmental SEM (ESEM) to strengthen its properties [6,7]. However, the macro properties of CMA are not as stable as hot-mix asphalt mixture due to a lack of research on reinforcement for the cement–emulsified asphalt binder (CA), which generally consists of emulsified asphalt and the proper quantity of cement.

However, apart from its microscopic characteristic, viscoelasticity is known to be critical to CA [8,9], and a dynamic shear rheometer is the most efficient way to capture the viscoelasticity of CA paste or composite binder [10]. For example, the influence of emulsified asphalt types on the viscoelasticity of CA and its influencing mechanism are researched from the indexes of apparent viscosity and yield stress [11], and then its viscoelasticity is predicted [12]. Furthermore, the fatigue properties and creep characteristics of CA composite materials were studied [13,14].

However, there have been no further studies on fiber-reinforced cement–emulsified asphalt binder, which, as a kind of composite binder, is essentially important for the CMA. Additionally, quantitative evaluation of the fiber reinforcement on CA based on viscoelasticity is necessary for maximizing the reinforcement effect. Therefore, in this study, two kinds of fibers are adopted as a reinforced phase in CA. Then, a quantitative evaluation of the reinforcement on creep recovery ability of fiber-reinforced CA is carried out based on the viscoelasticity by performing repeated creep recovery tests (RCRTs).

## 2. Materials and Experimental Methods

### 2.1. Materials

For the styrene–butadiene–styrene (SBS)-modified asphalt emulsion and P.O., 42.5 cement were selected for preparing the CA binder. Table 1 shows the basic technical indexes of emulsified asphalt. Table 2 shows the physical and mechanical properties of ordinary Portland cement.

According to previous research [15], polyester fiber (PF) and brucite fiber (BF) were adopted as reinforced phases in CA. Table 3 shows the basic technical characteristics and properties of polyester and brucite fibers, and Figure 1 shows the SEM images of the two fibers.

### 2.2. Specimen Preparation

According to previous research [10], the optimum ratio of cement and emulsified asphalt and the curing time was analyzed and determined. Therefore, in this paper, CA (i.e., control group) had a cement–residual asphalt ratio of 1:1 by mass; for composites with a fiber addition, the polyester fiber and brucite fiber content are 2%, 4%, and 6% (by mass of cement) by taking economic availability and dispersion into account [16].

Fiber-reinforced CA samples were prepared as follows. First, cement was gradually added to emulsified asphalt in a blender, and they were stirred well at low speed. Then, the mixes were stirred at high speed for 3 min by mechanical agitation. Thereafter, fibers were added to the mixes, and they were stirred at high speed for 3 min. Last, the mixes were placed into silicone molds and kept at room temperature (23 ± 2 °C) for curing. The demulsification of emulsified asphalt and the hydration of cement occurred especially in the first two days. Additionally, at the age of 7 days, the changes in viscoelasticity of CA became stable. Therefore, all the samples were tested after 7 days of curing [17]. The parallel tests for each group were performed, and the change regular of the parallel tests for each group were similar, so one set of experimental data was selected for analysis of the recovery rate of creep deformation, accumulated strain, and creep compliance.

### 2.3. Experimental Methods

Although multiple stress creep and recovery (MSCR) is a standardized test for determining the percent of recovery and non-recoverable creep compliance of asphalt binders, the repeated creep recovery test (RCRT) is also an effective method to evaluate the creep and recovery behavior of binder. Especially, RCRT can not only show the difference between recoverable and non-recoverable deformation as MSCR but also show the difference between instantaneous deformation recovery ability and delayed deformation recovery ability.

Repeated creep recovery tests (RCRT) were performed on a dynamic shear rheometer with parallel-plate geometry (25 mm diameter, 1 mm gap), shown in Figure 2. Because CA has an extremely high viscosity, the difference in creep deformations under low shear stress is small. Therefore, according to a previous study [10], the top plate applied shear stress of 1000 Pa, and the bottom plate was temperature-controlled, with a circulating water bath set to 60 °C during the whole test process. In order to measure the effect of fiber on the recovery rate of creep deformation and accumulated deformation, an RCRT protocol containing 100 creep-recovery cycles was applied according to references [18,19], shown in Figure 3a, in which creep time and recovery time were set at 1 s and 9 s, respectively. The sample initially deforms parabolically with the creep time, in which constant stress was applied for 1 s. During the recovery step, the shear stress was removed, and there was a certain recovery of deformation. Figure 3b shows a schematic of the evolution of the creep and recovery strain in cycles.

## 3. Results and Discussion

### 3.1. Recovery Rate of Creep Deformation

Herein, in order to quantitatively characterize the creep and recovery deformation in a single cycle, the creep-recovery ratio (CRR) is defined as the deformation recovery rate in every single cycle:(1)$$\mathrm{CRR}=\frac{D_{recovery}}{D_{creep}}\times 100\%$$
where *D_creep_* is the deformation in the 1 s creep process, and *D_recovery_* is the deformation recovery in the 9 s recovery process.

In the repeated creep recovery test (RCRT), after 50 cycles of loading, the development of the binder was considered to be stable, and the influence of delayed elasticity decreased. Therefore, the data of the 50th and 51st cycles of RCRT were usually used to analyze the creep-recovery deformation and fit the creep compliance [20].

In order to characterize the strengthening effect of fiber on CA composite, first, the creep deformation and recovery rate of fiber-reinforced CA with loading time were captured by RCRTs. Figure 4 shows the creep recovery curves of CA with polyester fiber in the 50th and 51st cycles. Compared to the plain CA, the creep recovery curves of CA with polyester fiber decrease significantly. In addition, when the polyester fiber content was 2% or 4%, the creep recovery curves showed little difference. However, for 6% polyester fiber-reinforced CA, the creep recovery curve experienced a significant decrease. Based on the creep deformation and CRR of CA with polyester fiber in the 50th and 51st cycles, shown in Table 4, the following conclusions were obtained:

After 49 cycles of creep and recovery, the accumulated strain of CA with polyester fiber was only about half or less than that of CA. It is apparent that fiber has a good strengthening effect on CA. Consequently, the deformation resistance of fiber-reinforced CA greatly increased.

The CRR of CA with polyester fiber in the 50th and 51st cycles, both of which were above 70%, exhibited obvious improvement over CA, which indicates that most of the strain generated by 1 s loading in each cycle can be recovered by itself, thereby having a positive effect on the deformation resistance and elastic recovery ability of the composites. In addition, it should be noted that the CRR of 4% polyester fiber-reinforced CA was slightly lower than that of composites with 2% polyester fiber in the 50th and 51st cycles, but the CRR of the composite with 6% polyester fiber exhibited a significant increase and was already more than 80%. The increase proves that there is an obvious improvement in the internal microstructure of the composite system when the polyester fiber content changes from 4% to 6%.

Similarly, the creep recovery curves of brucite fiber-reinforced CA in the 50th and 51st cycles are shown in Figure 5.

In comparison to Figure 4, some new characteristics appear in Figure 5. In the 50th and 51st cycles, the accumulated strains of CA with 4% and 6% brucite fiber were less than 1. Moreover, as shown in Table 5, the CRRs of CA with 2%, 4%, and 6% brucite fiber were 79.80%, 84.54%, and 93.31%, respectively, which are higher than those of polyester fiber-reinforced CA under the same conditions; they are also much higher than that of CA (i.e., 43.73%). The results show that brucite fiber-reinforced CA had better deformation recovery ability.

Unlike polyester fiber, when brucite fiber content changes from 2% to 4%, the creep recovery curve decreases greatly, but the decline is relatively small when the brucite fiber content is 4% to 6%. The difference between the two fibers is affected by the properties and distribution characteristics of the fiber. Specifically, the tensile strength of brucite fiber is greater than that of polyester; meanwhile, its diameter and length are less than polyester fiber. Therefore, brucite fibers are more likely to form bridging and have a reinforcing effect even if only small additions are made.

### 3.2. Accumulated Strain

Due to the viscoelastic characteristic of fiber-reinforced CA, a residual deformation occurs in each cycle. Therefore, the accumulated strain is the sum of the residual deformation of 100 cycles. It reflects the deformation resistance ability and deformation recovery ability of fiber-reinforced CA.

The above analysis was based on the creep process and recovery process in a single cycle. The following analysis focused on the accumulated strain development of fiber-reinforced CA in 100 creep-recovery cycles. Figure 6 shows the evolution of accumulated strains of polyester fiber-reinforced CA with time. In the beginning, the accumulated strain of the composite with 6% polyester fiber was even higher than that of the other two composites, and the accumulated strain of the composite with 4% polyester fiber was also greater than that of the composite with 2% polyester fiber content over a long period of time. The accumulated strain of the latter fiber content does not begin to exceed that of 4% fiber content until about the 50th cycle. In addition, similar to the accumulated strain curve of CA, the accumulated strain curves of the three polyester fiber-reinforced CAs increase linearly with time and the growth rates (i.e., the slope of the curve) decrease with the increase in fiber content. According to the final accumulated deformation value indicated in Figure 6, when 2%, 4%, and 6% polyester fibers were added to CA, the accumulated strain of the composites after 100 creep-recovery cycles decreased by 44.1%, 47.8%, and 60.9%, respectively. When the polyester fiber changes from 4% to 6%, the accumulated strain of polyester fiber-reinforced CA decreases significantly.

Figure 7 shows the evolution of accumulated strains of brucite fiber-reinforced CA with time. Compared to CA, the addition of brucite fiber slows down the linear growth process of the accumulated strain of CA. In addition, after 100 creep recovery cycles, the total accumulated strain is reduced by 57.9%, 88.6%, and 94.3%, respectively. Moreover, unlike the rule governing polyester fiber-reinforced CA, the accumulated strain decreases greatly when the brucite fiber content changes from 2% to 4%, while the decrease is relatively small when the brucite fiber content changes from 4% to 6%. This situation is consistent with the analysis in Figure 5 and Table 5.

According to the above analysis, the accumulated strains of the two fiber-reinforced CAs have common characteristics, such as the total accumulated strain being significantly reduced and the accumulated strain decreasing with the increase in fiber content. However, the fiber-reinforced CAs with different fibers also contain the following differences. At the same fiber dosage, the accumulated strain of brucite fiber-reinforced CA is smaller, and the creep deformation recovery ability stronger. This result can be tied to the properties of the fibers. As shown in Figure 1, polyester fibers are cylindrical bundles with a smooth surface, while brucite fibers are alkaline and consist of bundles or monofilaments with uneven thickness and rough surfaces. Therefore, although polyester fiber is more easily dispersed, brucite fiber has stronger adsorption to asphalt and cement hydration products. On the other hand, the tensile strength and other mechanical properties of brucite fiber are better than polyester fiber.

### 3.3. Creep Compliance

#### 3.3.1. The Improved Piecewise Curve-Fitting for the Creep Compliance

Based on the repeated creep recovery tests, the Burgers model was demonstrated to be applicable to fitting strain evolution of CA [10]. Its constitutive equation was applied as follows:(2)εt=σ0G0+σ0G11−e−tG1η1+σ0η0t
where *ε*(*t*) is strain and *σ*_0_ is a constant stress (Pa); *G*_0_ and *η*_0_ are the elastic modulus (Pa) and viscosity coefficient (Pa·s), respectively, of the Maxwell model; *G*_1_ and *η*_1_ are the elastic modulus (Pa) and viscosity coefficient (Pa·s), respectively, of the Kelvin model; and *t* is creep time (s).

Then, the constitutive equation of creep compliance was obtained by dividing both sides of Equation (2) with a constant stress *σ*_0_. Therefore, the creep compliance *J*(*t*), which is independent of the stress, reflects the real viscoelastic behavior of the material under load. Specifically, as seen in Equation (3), *J_E_* is instantaneous elastic deformation compliance, *J_C_* is delayed elastic deformation compliance, and *J_V_* is viscous flow deformation compliance. These three creep compliances indicate instantaneous elasticity, delayed elasticity, and viscous flow, respectively.
(3)Jt=εtσ0=1G0+1G11−e−tG1η1+tη0=JE+JC+JV

The most important step in calculating creep compliance is to find an accurate fitting method. For this study, an improved piecewise curve-fitting method was used to obtain the corresponding fitting parameters, which proceeds according to the following steps.

In the initial creep phase:

When *t→*0, ($1-e^{-\frac{G_{1}}{\eta_{1}}t})$ is an infinitesimal of a higher order than *t*, herein,
(4)$$\underset{t\to 0}{\lim}\frac{\left(1-e^{-\frac{G_{1}}{\eta_{1}}t}\right)}{t}=\underset{t\to 0}{\lim}\frac{G_{1}}{\eta_{1}}e^{-\frac{G_{1}}{\eta_{1}}t}=\frac{G_{1}}{\eta_{1}}$$
(5)$$\left(1-e^{-\frac{G_{1}}{\eta_{1}}t}\right)≅\frac{G_{1}}{\eta_{1}}t$$

Substituting Equation (5) into Equation (3) yields:(6)$$J\left(t\right)≅\frac{1}{G_{0}}+\left(\frac{1}{\eta_{0}}+\frac{1}{\eta_{1}}\right)t$$

When *t* is small enough, the parameters obtained by linear fitting are as follows:(7)$$\frac{1}{G_{0}}=a_{1}$$
(8)$$\frac{1}{\eta_{0}}+\frac{1}{\eta_{1}}=a_{2}$$

At the end of the creep phase, $\left(1-e^{-\frac{G_{1}}{\eta_{1}}t}\right)\to 1$, herein,
(9)$$\frac{1}{G_{1}}\left(1-e^{-\frac{G_{1}}{\eta_{1}}t}\right)≅\frac{1}{G_{1}}$$

Substituting Equation (9) into Equation (3) yields:(10)$$J\left(t\right)≅\left(\frac{1}{G_{0}}+\frac{1}{G_{1}}\right)+\frac{1}{\eta_{0}}t$$

When *t* is big enough, the parameters obtained by linear fitting are as follows:(11)$$\frac{1}{G_{0}}+\frac{1}{G_{1}}=b_{1}$$
(12)$$\frac{1}{\eta_{0}}=b_{2}$$

The Burgers model fitting parameters are obtained through Equations (7), (8), (11) and (12), and the fitting results of *G*_0_, *η*_0_, *G*_1,_ and *η*_1_ are presented in Table 6. Then the results of measured creep compliances in the 50th cycle and fitting curves are shown in Figure 8 and Figure 9. As can be seen from the figures, the improved piecewise curve-fitting method has high accuracy.

#### 3.3.2. The Reinforcement of Polyester Fiber on CA Based on Creep Compliance

By substituting the fitting parameters into Equation (3), the instantaneous elastic deformation compliance *J_E_*, delayed elastic deformation compliance *J_C_*, and viscous flow deformation compliance *J_V_* during the loading process are calculated. Lastly, the percentages of *J_E_*, *J_C_*, the sum of *J_E_* and *J_C_*, and *J_V_* in the total compliance (abbreviated as %*J_E_*, %*J_C_*, (%*J_E_* + %*J_C_*) and %*J_V_*) were plotted and analyzed.

Figure 10 shows the variations of %*J_E_*, %*J_C_*, (%*J_E_* + %*J_C_*), and %*J_V_* of polyester fiber-reinforced CA with loading time. In Figure 10a, the %*J_E_* of polyester fiber-reinforced CA falls off rapidly when loading time increases. Local magnification shows that most of the time, %*J_E_* of CA is the largest, while the %*J_E_* of CA with 6% polyester fiber is the smallest. Significantly, the %*J_E_* of CA with 4% polyester fiber is greater than that of the composite with 2% polyester fiber at the beginning, but then it becomes smaller than the latter over time.

As for the %*J_C_*, shown in Figure 10b, after an initial rise, all samples’ %*J_C_* decreases almost linearly, whereby the %*J_C_* of CA is the smallest, the %*J_C_* of CA with 6% polyester fiber is the largest, and the %*J_C_* of the composite with 4% polyester fiber gradually exceeds that of the composite with 2% polyester fiber. The change law of %*J_C_* is consistent with that of the sum of %*J_E_* and %*J_C_* (%*J_E_* + %*J_C_*) in Figure 10c. Meanwhile, the %*J_V_* of polyester fiber-reinforced CA increases significantly with loading time, which indicates that the longer the loading time, the easier it is for the non-recoverable deformation of the composite to accumulate. As the index of %*J_V_* indicates, the deformation recovery abilities of CA with different polyester fiber contents are significantly improved, and the deformation recovery ability of the composite with 6% polyester fiber is the strongest among the three fiber contents.

#### 3.3.3. The Reinforcement of Brucite Fiber on CA Based on Creep Compliance

Figure 11 shows the variations of %*J_E_*, %*J_C_* (%*J_E_* + %*J_C_*), and %*J_V_* of brucite fiber-reinforced CA with loading time. As can be seen from Figure 11a, the %*J_E_* of the composite with 2% brucite fiber is very close to that of CA in both the rapid decreasing stage and the stable stage. When brucite fiber content increases to 4% and 6%, the %*J_E_* of fiber-reinforced CA increases significantly. Therefore, the viscoelastic ratio of the composites is not changed when the brucite fiber content is 2%, but the instantaneous elastic recovery abilities of the composites are significantly enhanced when the brucite fiber content is 4% or 6%.

As for the %*J_C_*, shown in Figure 11b, with the addition of 2% brucite fiber, the improvement in %*J_C_* is noticeable in comparison to CA. However, a greater increase in the %*J_C_* is observed in CA with 4% and 6% brucite fiber. More specifically, the %*J_C_* of composite with 4% brucite fiber is greater than that of composite with 6% brucite fiber. Therefore, the (%*J_E_* + %*J_C_*) of CA with 4% brucite fiber is the largest, followed by the composite with 6% brucite fiber (Figure 11c). According to Figure 11c,d, 2% brucite fiber has a relatively small impact on the viscoelastic proportion of the composite; however, when the brucite fiber content increases to 4% and 6%, the composition proportion of creep compliance changes greatly. Nonetheless, the deformation recovery ability of the composite does not increase with the increase in brucite fiber content; the optimal content of brucite fiber is 4% in view of (%*J_E_* + %*J_C_*) and %*J_V_*.

## 4. Conclusions

This study investigated the fiber-reinforced effect in cement–emulsified asphalt binder. A repeated creep recovery test containing 100 creep-recovery cycles was implemented. CA with polyester and brucite fiber exhibit increased CRR compared to plain CA (43.73%), and the CRR of CA with 6% polyester and 6% brucite fiber increased to 81.23% and 93.31%, respectively, indicating that fiber enhances the deformation recovery ability. For the accumulated strain in the 100-cycle creep recovery process, CA with and without fiber exhibited a linear increase with cycles, wherein higher fiber content led to smaller accumulated strain, and brucite fiber was more effective in promoting deformation recovery (consistency with CRR).

The improved piecewise curve-fitting method is applicable to fitting creep compliance of fiber-reinforced CA and has high accuracy. Although the CA with 6% polyester fiber has the smallest %*J_E_*, it has the largest %*J_C_*. The (%*J_E_* + %*J_C_*) results show that the deformation recovery ability of CA is significantly improved with the addition of different polyester fiber dosages, and the polyester fiber-reinforced CA has the strongest deformation recovery ability with the addition of 6% fiber content. For the CA with brucite fiber, when the brucite fiber content changes from 2% to 4% or 6%, the viscoelastic proportion of the composite changes greatly. However, the elastic deformation recovery ability of the composite does not increase continuously with brucite fiber content; the optimal content is 4%. This different trend between the two fibers can be attributed to microtopography and mechanical properties, but these results confirm the reinforcement effect of fiber in CA.

## Figures and Tables

**Figure 1 materials-15-07451-f001:**
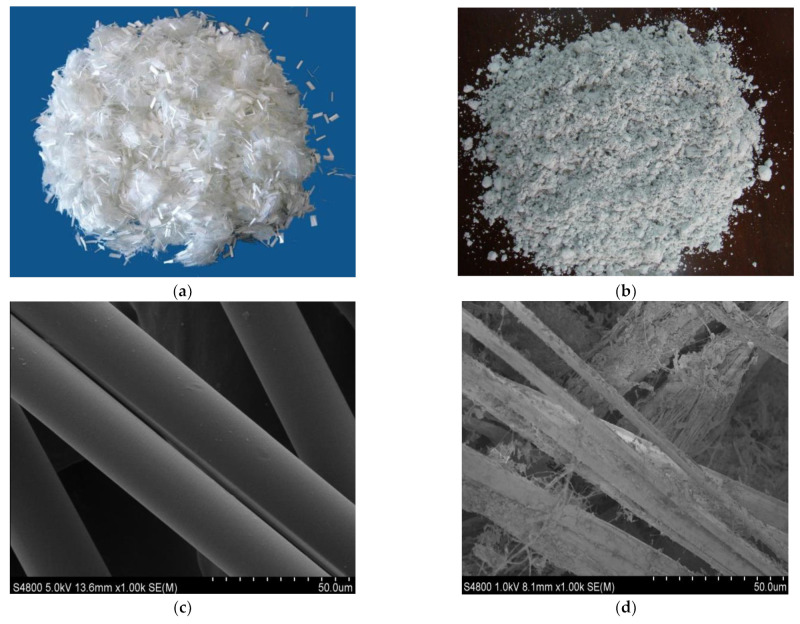
Photos and SEM images of (**a**,**c**) polyester fiber and (**b**,**d**) brucite fiber.

**Figure 2 materials-15-07451-f002:**
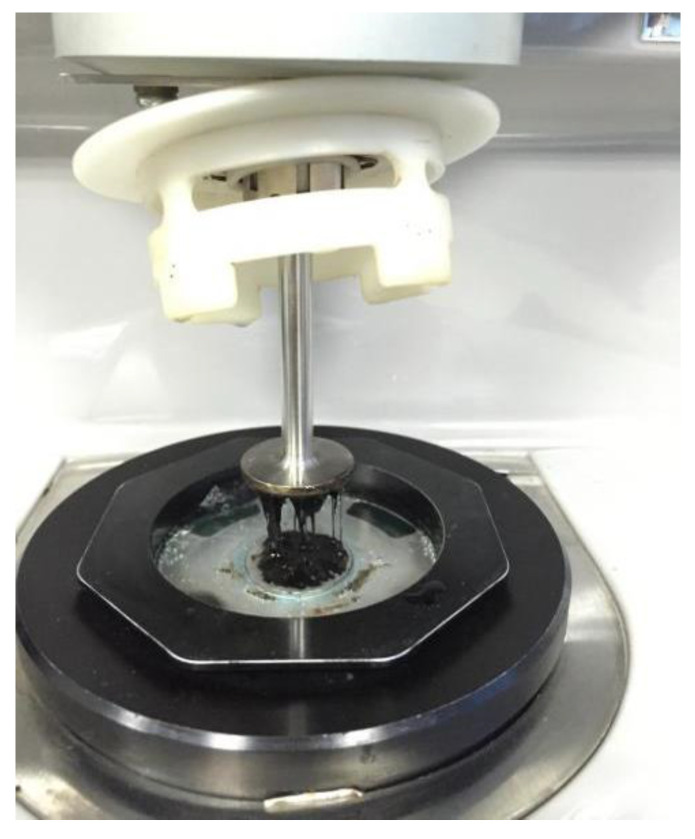
Fiber-reinforced CA samples after RCRT.

**Figure 3 materials-15-07451-f003:**
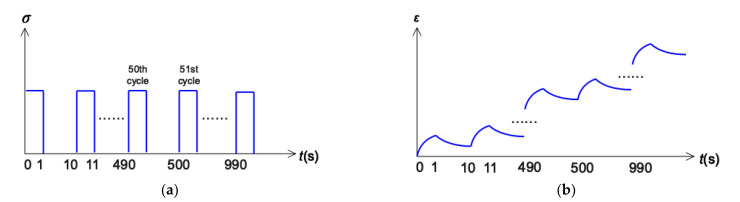
RCRT protocol: (**a**) stress and (**b**) accumulated strain.

**Figure 4 materials-15-07451-f004:**
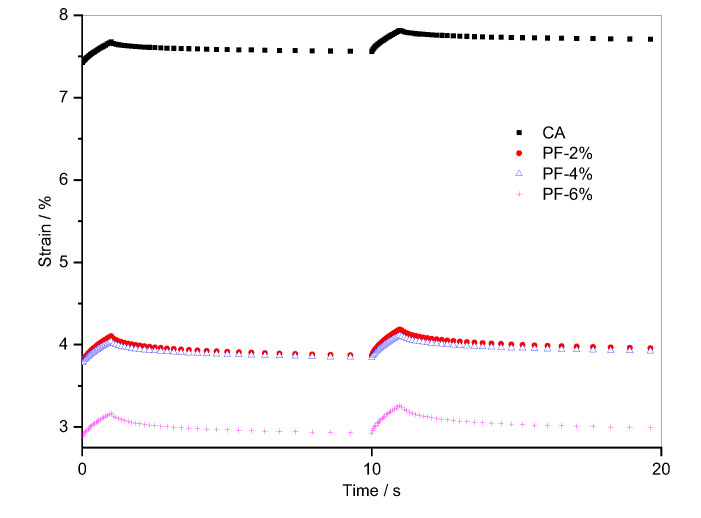
Evolution of strain versus time for composites with polyester fiber in the 50th and 51st cycles.

**Figure 5 materials-15-07451-f005:**
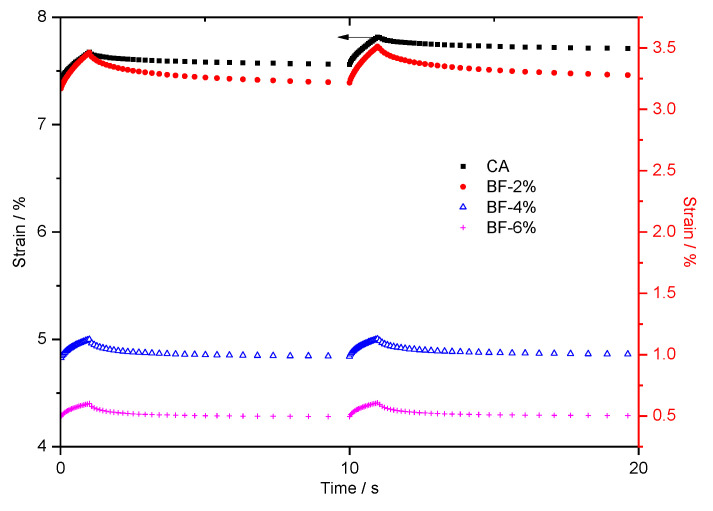
Evolution of strain versus time for composites with brucite fiber in the 50th and 51st cycles. (left *Y*-axis: CA, as indicated by the black arrow; right *Y*-axis: BF-2%, BF-4%, and BF-6%).

**Figure 6 materials-15-07451-f006:**
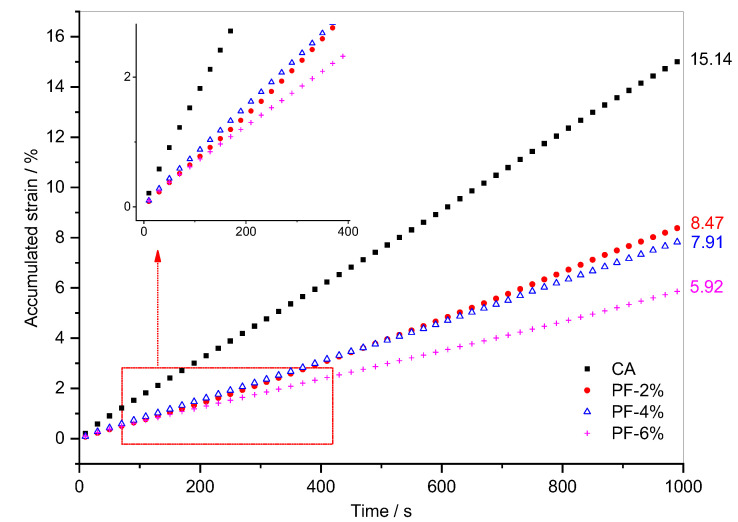
Evolution of accumulated strain versus time for composites with polyester fiber.

**Figure 7 materials-15-07451-f007:**
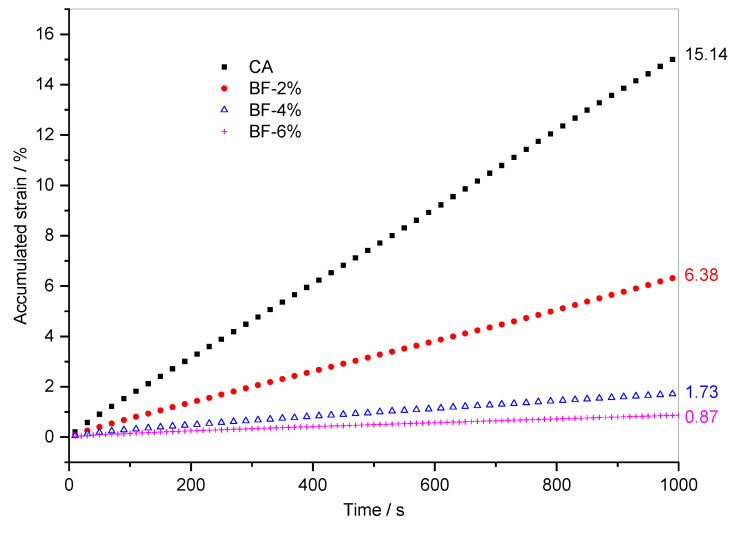
Evolution of accumulated strain versus time for composites with brucite fiber.

**Figure 8 materials-15-07451-f008:**
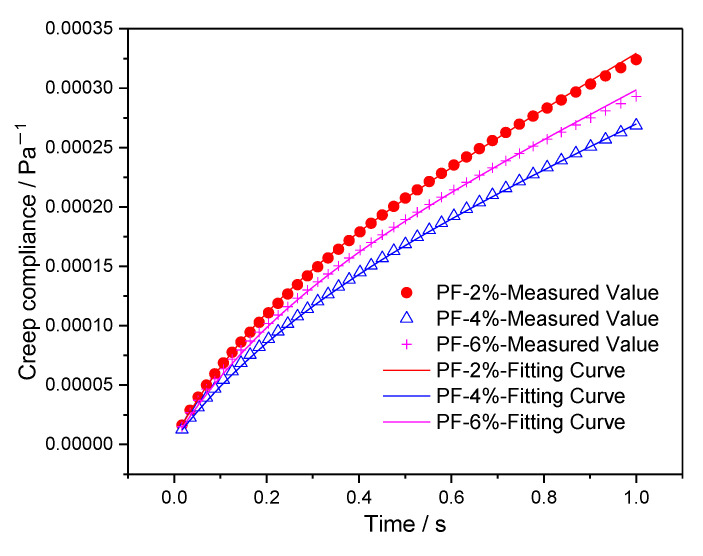
The measured and fitting results of creep compliance for composites with polyester fiber.

**Figure 9 materials-15-07451-f009:**
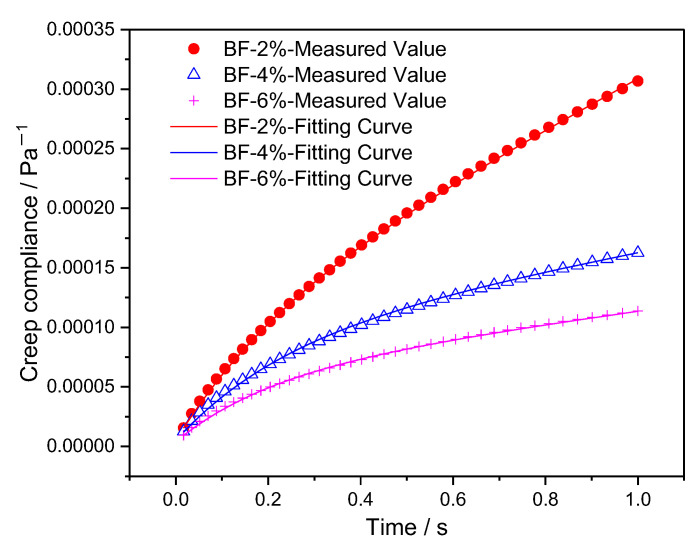
The measured and fitting results of creep compliance for composites with brucite fiber.

**Figure 10 materials-15-07451-f010:**
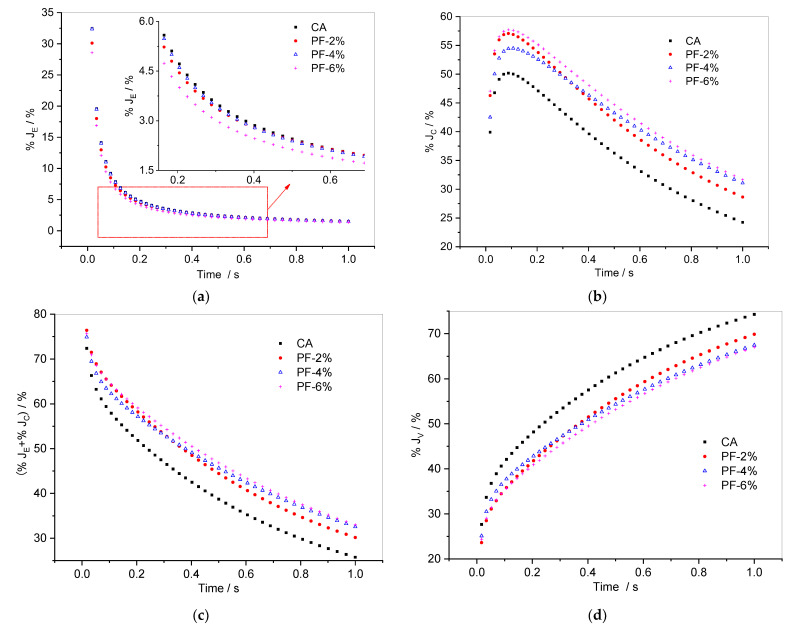
Evolution of (**a**) %*J_E_*, (**b**) %*J_C_*, (**c**) (%*J_E_* + %*J_C_*), and (**d**) %*J_V_* versus time for composites with polyester fiber in the 50th cycle.

**Figure 11 materials-15-07451-f011:**
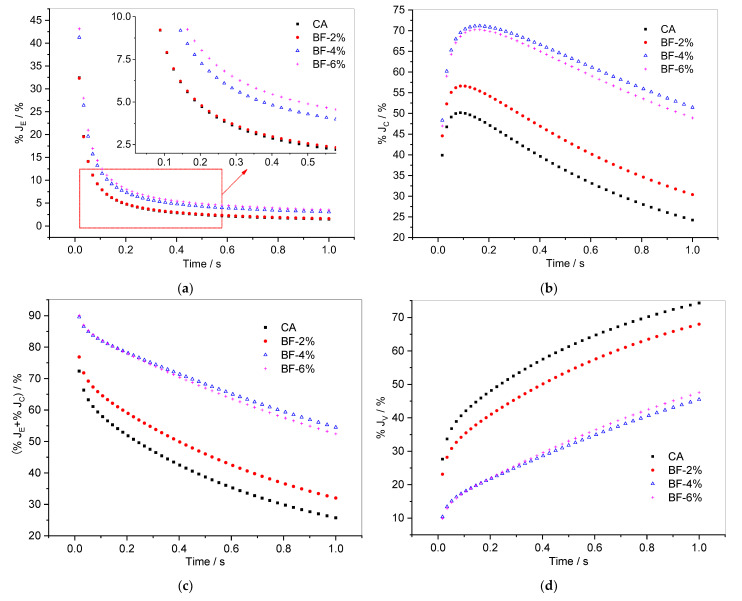
Evolution of (**a**) %*J_E_*, (**b**) %*J_C_*, (**c**) (%*J_E_* + %*J_C_*) and (**d**) %*J_V_* versus time for composites with brucite fiber in the 50th cycle.

**Table 1 materials-15-07451-t001:** Technical indexes of emulsified asphalt.

Items	Value
Viscosity (Saybolt–Furol, 25 °C)/S	25
Settlement (5 d)/%	1.2
1 d storage stability/%	0.1
Sieve test/%	0.02
Residue by distillation/%	61
*Residue tests*	
Penetration (25 °C, 100 g, 5 s)/0.1 mm	75
Softening point/°C	60
Ductility (25 °C, 5 cm/min)/cm	132
Ductility (5 °C, 5 cm/min)/cm	32
Solubility (trichloro ethylene)/%	99.5

**Table 2 materials-15-07451-t002:** Physical and mechanical properties of ordinary Portland cement.

Physical Properties	Value
Density/(g/cm^3^)	3.102
Specific surface/(m^2^/kg)	380
7-day comprehensive strength/MPa	20.9
28-day comprehensive strength/MPa	48.8

**Table 3 materials-15-07451-t003:** Basic technical characteristics and properties of the two fibers.

Items	Polyester Fiber	Brucite Fiber
Color and appearance	White, bunchy	Hoary, flocculent
Average length/mm	4–6	0.2–4
Average diameter/μm	16–20	2–4
pH	7.0	9.0
Density/(g/cm^3^)	1.318	2.286
Oil absorption rate/%	4.25	3.62
Moisture absorption rate/%	2.23	1.08
Tensile strength/MPa	580	900

**Table 4 materials-15-07451-t004:** Creep deformation and CRR of polyester fiber-reinforced CA in the 50th and 51st cycles.

No.	Items	CA	PF-2%	PF-4%	PF-6%
50th cycle	initial strain	7.413	3.781	3.768	2.873
	strain at the end of creep	7.676	4.105	4.037	3.166
	strain at the end of recovery	7.561	3.868	3.843	2.928
	CRR_50_/%	43.73	73.15	72.12	81.23
51st cycle	initial strain	7.561	3.868	3.843	2.928
	strain at the end of creep	7.822	4.193	4.118	3.266
	strain at the end of recovery	7.707	3.955	3.919	2.991
	CRR_51_/%	44.06	73.23	72.36	81.36

**Table 5 materials-15-07451-t005:** Creep deformation and CRR of brucite fiber-reinforced CA in the 50th and 51st cycles.

No.	Items	CA	BF-2%	BF-4%	BF-6%
50th cycle	initial strain	7.413	3.155	0.962	0.488
	strain at the end of creep	7.676	3.462	1.124	0.601
	strain at the end of recovery	7.561	3.217	0.987	0.495
	CRR_50_/%	43.73	79.80	84.54	93.31
51st cycle	initial strain	7.561	3.217	0.987	0.495
	strain at the end of creep	7.822	3.517	1.130	0.609
	strain at the end of recovery	7.707	3.277	1.003	0.503
	CRR_51_/%	44.06	80.00	88.63	93.23

**Table 6 materials-15-07451-t006:** The fitting parameters of Burgers model.

	**Parameters**	***G*_0_/Pa**	***η*_0_/(Pa·s)**	***G*_1_/Pa**	***η*_1_/(Pa·s)**
**No.**					
CA		250,000.0	5000.0	15,151.5	3333.3
PF-2%		200,000.0	4347.8	10,526.3	2127.7
PF-4%		250,000.0	5494.5	11,627.9	3144.7
PF-6%		250,000.0	5000.0	10,416.7	2500.0
BF-2%		200,000.0	4761.9	10,526.3	2381.0
BF-4%		200,000.0	13,513.5	11,764.7	2809.0
BF-6%		250,000.0	18,518.5	17,857.1	3759.4

## Data Availability

The raw and processed data required to reproduce these results are available by contacting the authors.

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
