# Peer review of "Effect of Fiber Reinforcement on Creep and Recovery Behavior of Cement–Emulsified Asphalt Binder"

_materials, 2022, doi:10.3390/ma15217451_

Round 1
Reviewer 1 Report
The authors present an interesting study on fiber reinforcement of cement-emulsified asphalt binders using creep recovery tests in the dynamic shear rheometer. The structure and methodology are clear and results are extensively analysed. However, I have some general concerns about the eligibility of this manuscript for publication and have the following comments for major revision.
- Title is grammatically incorrect, misleading and confusing. I propose to change the title to something like “Effect of fiber reinforcement on creep and recovery behavior of cement-emulsified asphalt binder”.
- Use of the term “asphalt” is not consistent within the manuscript and should be unified. Please always differentiate between “asphalt binder” and “asphalt mixture” and specify accordingly. In this context it is also unclear if the acronym “CA” is used for cement-emulsified asphalt binder or for cement-emulsified asphalt mixture. Please clarify throughout the document.
- One of my main concerns is about the creep and recovery test chosen. Please clarify and explain why you chose this arbitrary number of 100 cycles and a shear stress of 1 kPa? Are there any previous publications or standards for this test? Why did you not apply the well known Multiple Stress Creep and Recovery Test (MSCR), which is standardized for this kind of application in AASTHO, ASTM and EN standards?
- The extent of the testing performed is very limited. You only have 7 material variants where you performed one test of 1000 s each. This is only one day of work and rather limited for a scientific publication. I recommend that you at least perform one more replicate for each test, this might already explain some of the inconsistencies identified.
- L 58: Here you mention that you prepared CA mastics. However, in the remaining part of the paper these mixtures are only titled as “CA”. Please clarify if the material you tested should be named as “cement-emulsified asphalt binder” or “cement-emulsified asphalt mastic” and use the terms consistently (see also second comment).
- L 79: Please explain the term silastic. Are this silicone molds? What dimensions do they have? What exactly happens during the 7-day curing?
- L 82: it is “dynamic” shear rheometer and not “dynamical”.
- L 101: should be CRR instead of CCR
- Why did you not use well known-parameters for analysis of the creep and recovery test? Typically the non-recoverable creep compliance JNR and the percent recovery R are used, for example in the previously mentioned MSCR test. This would allow the readers to better understand and compare the results.
- Why did you chose the arbitrary values of 50th and 51st cycles for extended analysis? Shouldn’t you be better analyze all the cycles? Usually mean values of creep compliance and recovery are reported for such kind of tests.
- Figure 5: why do you use two separate Y-axis and which data belongs to which axis?
- L 226: I think this should be G0 instead of E0
- Figure 8 and Figure 9: Is this the first cycle? Please clarify.
- I did not understand the piecewise fitting. Why did you use such many simplifications and approximations? Usually you can just determine model parameters for Burgers model by least squares approximation. Are there any references for this method? Please revise?
- Instead of showing the %J values it would be interesting to actually present and compare the model parameters. Are the model parameters constant for all 100 load cycles? Can the model also approximate the recovery phase? How are the model parameters affected by the fibers?
- In the abstract you say, that Burgers model and piecewise curve-fitting is used. However, in my understanding, you used a piecewise curve-fitting to determine the Burgers model parameter. Please clarify and revise.
Author Response
Firstly, we want to express our gratitude to the two reviewers and editors for great help. Followings are our detailed response about reviewers’ comments to our manuscript (ID: materials-1890287, The reinforcement of fiber on the cement-emulsified asphalt based on viscoelasticity).
Reviewer #1:
- Thanks for the kindly suggestions. We havechanged the title to “Effect of fiber reinforcement on creep and recovery behavior of cement-emulsified asphalt binder”.
- In the revised manuscript, wehaveclarified that CA indicated cement-emulsified asphalt binder. And we have revised some “asphalt” to “asphalt binder” or “asphalt mixture” for consistency.
- Thanks for the kindly suggestion. MSCR testisa standardized test for determining the percent of recovery and non-recoverable creep compliance of asphalt binders. Repeated creep recovery test (RCRT) is also an effective method to evaluate the creep and recovery behavior of binder. In previous publications, such as “Bahia H.U., Hanson D.I., Zeng M., et al. Characterization of modified asphalt binders in Superpave mix design[R]. Washington: National Cooperative Highway Research Program, 2001” “Mauricio Reyes, Igor B. Kazatchkov, Jiri Stastna, et al. Modeling of repeated creep and recovery experiments in asphalt binders[J]. Transportaion research board, 2009, 1:63-72”, “100 cycles” is the most commonly used parameter. And the shear stress varies with the binder, 1000Pa is determined based on our previous research (reference 10).
- Inthismanuscript, there only have 7 materials. But we did perform parallel tests for each group, and the change regular of the parallel tests for each group were similar. Therefore, we selected one set of experimental data for analysis.
Besides, we also tried to use different shear stress (100Pa). However, because CA had extremely high viscosity, the difference of creep deformations under this shear stress was small. Therefore, we adopted 1000Pa as the shear stress in this manuscript.
- We have revised the “mastics” to “binder” in Line 58.
- Line 79: It was silicone mold, its dimensions was25 mm indiameter, 1 mm in depth.
During the 7 days’s curing, the demulsification of emulsified asphalt and the hydration of cement occurred especially in the first two days. And at the age of 7days, the changes of viscoelasticity became stable. This had been discussed in our previous study (“Time-dependent properties of the viscoelasticity of cement-emulsified asphalt composites”)
- Wehaverevised the “dynamical” to “dynamic”.
- Wehaverevised the “CCR” to “CRR”.
- Thanks for the kindly suggestion. MSCR testandnon-recoverable creep compliance allow the readers to better understand. In this manuscript, the creep compliance J(t) was divided into three parts: instantaneous elastic deformation compliance JE, delayed elastic deformation compliance JC and viscous flow deformation compliance JV. These three creep compliances indicated instantaneous elasticity, delayed elasticity, and viscous flow, respectively. Therefore, this method could not only show the difference between recoverable and non-recoverable deformation, but also show the difference between instantaneous deformation recover ability and delayed deformation recover ability. And from fig.10 and fig.11, the delayed deformation recover ability (i.e. %JC) accounted for a large proportion.
- Inrepeated creep recovery test (RCRT), after50 cycles’ loading, the development of binder was considered to be stable, and the influence of delayed elasticity decreased. Therefore, the data of 50th and 51st cycles were usually used to fit the creep compliance in RCRT. It was also suggested by some previous publications, for example, “High temperature index of asphalt based on repeated creep[J]. Journal of South China University of Technology (Natural Science Edition)”
- Becausethestrain of CA and the strains of CAs with 2%, 4% and 6% brucite fiber varied widely, if only one Y-axis was used, it was difficult to show the differences between the strains of CAs with 2%, 4% and 6% brucite fiber. The strain of CA belonged to left Y-axis (marked by a black arrow), and the strains of CAs with 2%, 4% and 6% brucite fiber belonged to right Y-axis. We have added an instruction in the revised manuscript.
- Line226: Yes, it is G0not E0. We have corrected this mistake. Thanks.
- Figure 8 and Figure 9: These data were the measured creep compliances in the 50th cycle. We have clarified in the revisedmanuscript.
- Therearemany methods to determine model parameters for Burgers model. The improved piecewise curve-fitting method used in the manuscript had high accuracy. It could be seen from the Figure 8 and Figure 9, and from previous research (“DSR test of the visco-elasticity of asphalt mastics[D]. Zhengzhou University, 2007”).
- According to the suggestion, we have added the fitting model parameters in table 6. However this model can not approximate the recovery phase. In addition, the calculationofthe percentage of JE, JC and JV can reflect the instantaneous deformation recover ability, delayed deformation recover ability and non-recoverable deformation, but these model parameters do not. That is why we analyze %JE, %JC, (%JE+%JC) and %JV.
- We have revised this sentence in abstract accordingtothe reviewer's suggestion. Thanks again.
Reviewer 2 Report
Dear authors
I enjoyed reading your article, but some questions seemed unclear and needed further clarification.
-line11[ (2%, 4%, and 6% addition by mass of cement)],What is the reason for selecting these percentages? Please add the reference.
-In line75, How was the method of mixing and how to ensure homogeneous mixing?, please add more explanations.
-In line 69, Please bring two images of the fibers used before mixing with bitumen.
-In line 79, according to which standard were these time and temperature selected? Please provide a reference.
-In line 135 in Figure 5, what is the difference between the vertical axis on the left and the right? Please make it clearer in the figure.
-In the graph of Figure 6, why are the curves still upward with a constant slope?
-Scientific references should also be written in line 220 and the following equations.
-In Figures 8 and ... to the end, the curve of various changes has been drawn and the conditions that happened in the sample of fibrous bitumen have been explained, but there are few logical and scientific reasons and supporting references, please add references and reasons.
-In general, which fibers are more suitable and which percentage is better, so that they can be mixed together? Please discuss it
-What is the difference between this study and the following research?
Effect of Aging on the Viscoelastic Mechanical Properties of Cement/Emulsified Asphalt Composite Repair Material/ DOI:10.1520/JTE20220024
With respect.
Author Response
Firstly, we want to express our gratitude to the two reviewers and editors for great help. Followings are our detailed response about reviewers’ comments to our manuscript (ID: materials-1890287, The reinforcement of fiber on the cement-emulsified asphalt based on viscoelasticity).
Reviewer #2:
- Whenthe fiber addition exceeded 6%, it was very difficult to mix, while the fiber reinforcement was not obvious when the fiber is very low. Moreover, in our previous publication, the approximate fiber dosage range had been discussed from the properties of fiber-reinforced cement emulsified asphalt mixture. We have added the reference in the revised manuscript.
- Themethod of mixing was mechanical agitation. We have added this information in the revised manuscript.
In order to ensure homogeneous mixing, longer period of time was chosen for stirring.
- Wehave added two images of the fibers used before mixing with bitumen.
- The mixing and stirring time of CA as well as the curing temperature were selected based on our previous study, we have added the reference in the revised manuscript. While the mixing and stirring time of CA with fiber were determined based on the dispersion of fiber in
- Thestrain of CA belonged to left Y-axis (marked by a black arrow), and the strains of CAs with 2%, 4% and 6% brucite fiber belonged to right Y-axis. We have added an instruction in the revised manuscript.
- In Figure 6, the points represented the accumulated strain at the end of each creep-recovery cycle. The test results showed that the accumulated strain increase linearly with creep-recovery cycle.
- Equations(3)~(12) were deduced based on mathematics, so there was no reference.
- Figure 8 and Figure 9 illustrated the measured creep compliances and fitting curves. Thepurpose of this two pictures was to prove the high accuracy of improved piecewise curve-fitting method.
- This study focusedmainly on the reinforcement effect of the two fibers with different contents. As we explained in the conclusion, the polyester fiber-reinforced CA has the strongest deformation recovery ability with the addition of 6% fiber content. For the CA with brucite fiber, the optimal content is 4%. However, we didn’t discuss which fiber was more suitable. Future research could try to find which fibers are more suitable. Thanks for the suggest
- Thementioned research (Effect of aging on the viscoelastic mechanical properties of cement/emulsified asphalt composite repair material) is a quite new and significative paper. The common point of this two researches is that the creep behavior of cement-emulsified asphalt is analyzed by dynamic shear rheometer. However, the major differences between the two are the following. First, the reinforcement effect of two fibers were analyzed in our manuscript, however, in the above research, only cement/emulsified asphalt was researched. Second, in our manuscript the creep compliance was fitted, then the instantaneous elastic deformation compliance, delayed elastic deformation compliance and viscous flow deformation compliance were calculated and analyzed comparatively.
However, the mentioned research evaluated the influence of aging, which can provide reference for our future research. Thanks for the suggestion.
Thanks again for the valuable suggestions and kind help of reviewers and editors.
Round 2
Reviewer 1 Report
Dear authors,
while the answers to my comments and suggestions are satisfactorily, the changes in the manuscript are not sufficient. Please include all the information you provided in the answers also in the manuscript. It is necessary to improve the manuscript in order for other readers to better understand the methodology.
Explicitly, I would like to see in the revised paper:
- why to choose specific test parameters 100 cycles and 1 kPa
- why only present this seven materials
- why to choose cycle 50 and 51 for analysis
- why to choose 7 days curing time
- why not use the MSCR test (this is actually the successor of the RCRT)?
- how do the model parameters change over 100 cycles? Is there any equilibrium?
(See my previous comments for details)
Author Response
Thanks for the reviewers’ recognition. Followings are our detailed response about reviewers’ comments to our manuscript (ID: materials-1890287, The reinforcement of fiber on the cement-emulsified asphalt based on viscoelasticity).
Reviewer #1:
- We have explained why we choose the test parameters 100 cycles and 1 kPa and added references in revised manuscript (line 106; line 100-102).
- We have explained why we only present this seven materials (line 88-90).
- We have explained why we choose cycle 50 and 51 for analysis and added references (line 125-128).
- Wehave explained why we choose 7 days curing time and added references (line 85-88).
- Wehave explained why we use the RCRT test (line 92-98).
- Inour manuscript and previous studies, the data of 50th and 51st cycles of RCRT were used to fit the creep compliance, then the Burgers model parameters were calculated. Although whether the model parameters change with cycles and whether there is equilibrium of the model parameters are interesting points, they were not discussed in our manuscript and previous studies. The detailed reasons are as follows: First, that is lot of work to calculate parameters of 100 cycles. Second, according to referees, the creep compliance development of binder is considered to be unstable in early cycles, so there is not much point in analyzing the change of parameters with cycles. In spite of this, according to the reviewer's suggestion, we checked the test data of creep compliance of each cycle, the results show that, as the creep-recovery cycle increases, the changes of creep compliance with loading time are becoming more similar.
Thanks again for the valuable suggestions and kind help of reviewers and editors.
Reviewer 2 Report
Dear authors
Thanks for the corrections
Wishing you success
Author Response
Thanks for the reviewer’s approval.